# A Gas Flow Measurement System Based on Lead Zirconate Titanate Piezoelectric Micromachined Ultrasonic Transducer

**DOI:** 10.3390/mi15010045

**Published:** 2023-12-25

**Authors:** Tao Liu, Zhihao Li, Jiahuan Zhang, Dongxiao Li, Hanjie Dou, Pengfan Wu, Jiaqian Yang, Wangyang Zhang, Xiaojing Mu

**Affiliations:** Key Laboratory of Optoelectronic Technology and Systems of Ministry of Education, International Research and Development Center of Micro-Nano Systems and New Materials Technology, Chongqing University, Chongqing 400044, China; lt1903039473@163.com (T.L.); 202208131131@stu.cqu.edu.cn (Z.L.); woai1997rango@163.com (J.Z.); dongxiao@nus.edu.sg (D.L.); hanjiedou@cqu.edu.cn (H.D.); pengfanwu@cqu.edu.cn (P.W.); jiaqian_yang0517@163.com (J.Y.); zhangwangyang361@163.com (W.Z.)

**Keywords:** MEMS, ultrasonic flowmeter, PMUTs array, PZT

## Abstract

Ultrasonic flowmeter is one of the most widely used devices in flow measurement. Traditional bulk piezoelectric ceramic transducers restrict their application to small pipe diameters. In this paper, we propose an ultrasonic gas flowmeter based on a PZT piezoelectric micromachined ultrasonic transducer (PMUT) array. Two PMUT arrays with a resonant frequency of 125 kHz are used as the sensitive elements of the ultrasonic gas flowmeter to realize alternate transmission and reception of ultrasonic signals. The sensor contains 5 × 5 circular elements with a size of 3.7 × 3.7 mm^2^. An FPGA with a resolution of ns is used to process the received signal, and a flow system with overlapping acoustic paths and flow paths is designed. Compared with traditional measurement methods, the sensitivity is greatly improved. The flow system achieves high-precision measurement of gas flow in a 20 mm pipe diameter. The flow measurement range is 0.5–7 m/s and the relative error of correction is within 4%.

## 1. Introduction

Flow is one of the important measurement parameters in modern production and life. In people’s daily lives and industrial production activities, flow detection technology is an interesting research topic [1]. According to different design principles and basic structures, flowmeters are generally divided into three categories: mechanical type [2], electromagnetic type [3], and ultrasonic type [4]. Among them, ultrasonic flowmeters are characterized by high precision, high response speed, and high flow rate. Advantages such as reliability and non-contact measurement [5] gradually win out, attracting a large number of usage requirements. Coupled with the vigorous development of semiconductor technology, electronic technology, and MEMS technology in recent years, ultrasonic flowmeters are gradually developing towards miniaturization, integration, and intelligence [6,7,8,9].

The core component of an ultrasonic flowmeter is an ultrasonic transducer. The ultrasonic transducer can receive ultrasonic signals and convert them into electrical signals, or convert electrical signals into ultrasonic signals and transmit them to the external medium. Related research on ultrasonic transducers is promoting the development of the ultrasonic testing industry. Most of the ultrasonic transducers used in traditional ultrasonic flowmeters are body piezoelectric transducers [10,11,12,13]. Due to limitations of manufacturing technology, currently, commercially available body piezoelectric ultrasonic transducers are relatively large; therefore, there are limitations. With the development of MEMS technology, silicon-based MEMS processing technology, and MEMS-sensitive film material preparation technology, micromechanical ultrasonic transducers have been widely used in various fields. Compared with traditional transducers, MUT has the characteristics of small size and high integration. According to the working mechanism, MUTs have two types—piezoelectric MUTs (PMUTs) and capacitive MUTs (CMUTs). Compared with CMUTs, PMUTs have higher capacitance, which means that the effect of system parasitic capacitance on coupling and sensitivity is not as pronounced as CMUTs. Moreover, PMUTs have lower electrical impedance, which can better match the supporting electronic circuits and lower sensitivity parasitic capacitance [14,15]. Due to the different working mechanisms, PMUTs do not require high DC bias voltages such as CMUTs [16]. In addition, PMUTs do not require the fabrication of extremely small gaps [17] under curved films in order to achieve satisfactory sensitivity. The abovementioned advantages of PMUTs reduce the susceptibility to fabrication precision and array inconsistency, which greatly simplifies the complexity of the system [18]. PMUTs have successfully been demonstrated in several applications such as rangefinders [10], solid defect detection [19], ultrasonic imaging [20], and many other fields. PMUTs containing thin films based on AlN, ScAlN, and ZnO have been widely studied, but they all have the disadvantage of low-emission sound pressure. In this paper, a single crystal PZT film based on the magnetron sputtering method was developed, which has the advantages of large emission sound pressure. Compared with the PZT film processed using the sol–gel method, the magnetron sputtering method has a higher film-forming speed and process controllability, which can realize rapid and accurate film growth and produce a dense and high crystallite film structure. This method is suitable for a variety of substrates. Moreover, the thickness, composition, and microstructure of the film can be easily controlled, and the film is easy to produce in large sizes, which provides an effective process choice for the preparation of high-performance thin film devices. In previous reports, P.L.M.J. van Neer et al. [21] studied and verified the feasibility of PMUT-based gas flowmeters. In this study, the team first matched the resonant frequencies of the transmitting and receiving PMUTs by applying DC bias voltages to the transmitting end and receiving end, respectively, and then installed the matched pair of PMUTs on the inner wall of the pipe for testing. Zhu et al. [22] proposed an ultrasonic liquid flowmeter based on AlN thin film PMUT arrays. Two 10 × 10 PMUT arrays were installed outside the measured pipeline to measure the liquid flow in the pipeline without contacting the liquid. Ding et al. [23] proposed a blood flowmeter using the pulse Doppler ultrasonic detection method based on an AlN thin film PMUT for real-time monitoring of pulsatile blood flow and flow direction. Xiu et al. [24] developed a gas flowmeter using a single PMUT array. The PMUT array elements on a single array were grouped into transmitters and receivers, and a complete flow sensor was constructed using the cross-correlation method and cosine interpolation method. The system reflects changes in flow rate through the recorded TOF. Gao et al. [25] designed an ultrasonic flowmeter based on ScAlN PMUTs, which is suitable for small-diameter pipe applications and can be used for pipes with a diameter of 2 mm.

In this paper, a small pipe diameter flow measurement system based on a PZT PMUT ultrasonic transducer is proposed, which realizes the Magnetron sputtering preparation of a single-crystal PZT thin film PMUT, and increases the length of sound channel through the design of pipe shape, so as to effectively improve the measurement accuracy. The two PMUT arrays are mounted outside the tube wall and are therefore isolated from the gas under measurement. In order to improve the receiving performance of the gas flow measurement system, a signal conditioning circuit at the ultrasonic receiving end is designed, including a charge amplification circuit, a bandpass filter circuit, and an analog-to-digital conversion circuit. In order to meet the frequency, amplitude, and other requirements of the PMUT device for the excitation signal. The signal generator and transmitting circuit of the ultrasonic transmitting end are designed. In order to achieve high-precision measurement of ultrasonic time-of-flight difference, the FPGA peripheral circuit and internal measurement logic are designed. Finally, the ultrasonic time-of-flight test system verified that the PMUT device has good transceiver performance. In the DN 20 pipeline, the designed flow measurement system achieved flow detection of 0.5–7 m/s. In the measurement results, the corrected relative error was about 4%.

## 2. Measurement Principle

If a pair of ultrasonic transducers are placed opposite each other and their center line is not perpendicular to the flow velocity of the fluid medium, then during the propagation of ultrasonic waves, its propagation speed will be affected by the component of the fluid flow velocity in the direction of ultrasonic wave propagation [26]. When the propagation direction of ultrasonic waves is consistent with the direction of the fluid flow velocity component, its propagation speed is increased, and the ultrasonic wave flight time from transmission to reception is shortened. Accordingly, when the ultrasonic wave propagation direction is opposite to the direction of the fluid flow velocity component, its propagation speed is reduced and the ultrasonic time of flight from transmission to reception becomes longer, forming a downstream countercurrent ultrasonic flight time difference. Based on the condition that the propagation speed of ultrasonic waves in the static medium in this environment is known, the current flow rate of the fluid can be calculated through this time difference. Thus, the flow rate of the pipe is obtained.

The schematic diagram of flow measurement using the transit-time method is shown in Figure 1a. In the pipeline under test, ultrasonic transducer A and ultrasonic transducer B are installed on both sides of the pipeline at an oblique angle. This structure can be used to measure gases [27] and liquids [28]. The center distance between the two transducers is L, and the center line between the two intersects with the central axis of the pipe. The angle is *θ*. The pipe is filled with a gas medium with a flow rate of *v*. The velocity direction is shown in the figure. Next, define the direction in which the sound wave is emitted from ultrasonic transducer A to ultrasonic transducer B as the downstream direction, and record the ultrasonic time of flight as *t*_down_; the direction in which the sound wave is emitted from ultrasonic transducer B to ultrasonic transducer A is the countercurrent direction. Finally, the ultrasonic time of flight can be recorded as *t*_up_.

In Figure 1a, the component of the flow velocity of the medium in the pipeline in the direction of the line connecting the centers of the two transducers is *v*cos*θ*. After the sound velocity components are superimposed, the ultrasonic signal propagation speed along and against the flow and the counter flow propagation speed are, respectively:(1)$$c_{\mathrm{down}}=c_{0}+v\cos\theta$$
(2)$$c_{\mathrm{up}}=c_{0}-v\cos\theta$$

In the formula, *c*_0_ is the propagation speed of ultrasonic waves without medium flow, and *c*_down_ and *c*_up_ represent the propagation speed of downstream sound waves and the propagation speed of countercurrent sound waves, respectively. Based on this, the sound wave propagation time during downstream and countercurrent can be obtained:
(3)$$t_{\mathrm{down}}=\frac{L}{c_{0}+v\cos\theta }$$
(4)$$t_{\mathrm{up}}=\frac{L}{c_{0}-v\cos\theta }$$

From Equations (3) and (4) we can obtain:(5)$$v=\frac{L}{2\cos\theta }\left(\frac{t_{\mathrm{up}}-t_{\mathrm{down}}}{t_{\mathrm{up}}t_{\mathrm{down}}}\right)$$

Using the time difference method, the fluid flow velocity *v* can be measured by simply obtaining *t*_up_ and *t*_down_. After obtaining the fluid flow velocity, the instantaneous flow rate of the fluid can be calculated based on the diameter of the measured pipe:(6)$$Q=v\pi {\left(\frac{D}{2}\right)}^{2}$$

Among them, *D* is the diameter of the pipe in the measurement section, *Q* is the instantaneous flow rate of the fluid in the pipe, and the unit is m^3^/s.

The test accuracy of traditional flow detection methods will be affected by the diameter of the pipe. As the pipe diameter becomes smaller, the acoustic path of the transducer becomes shorter, placing higher accuracy requirements on the signal processing hardware circuitry. In order to extend the propagation path of ultrasonic waves and thereby improve the sensitivity of the gas flowmeter and the detection accuracy of TOF, a flowmeter structure is used to measure liquids [29], as shown in Figure 1b. Installing two PMUTs at both ends of the pipe to be measured allows ultrasonic waves to propagate directly along the pipe where the gas flows, increasing the difference between the downstream propagation time and the countercurrent propagation time, thus improving the sensitivity of the flowmeter. At the same time, since the acoustic path is not limited by the diameter of the pipe, appropriately increasing the length of the acoustic path can effectively improve the sensitivity of flow measurement to meet different application requirements. At the same time, this method is also more suitable for measuring fluid flow in small pipes.

## 3. The Design of the PMUT Array

### 3.1. Acoustic Transducer Design and Fabrication

In order to improve the performance of the PMUT and achieve better transmitting and receiving effects, a PMUT array is designed. The PMUT structure is shown in Figure 2. The PMUT device consists of 25 PMUT vibration elements, and all vibration elements form a 5 × 5 PMUT array; the top electrodes of the 25 vibrating elements are connected together and lead out as the top electrode of the device. The bottom electrode of the device is connected to the Pt electrode material under the PZT piezoelectric layer. The substrate of the device is an SOI wafer, and a cavity is formed on the back of the device via etching. When an alternating current signal of a certain amplitude and frequency is applied to the PMUT device through the top electrode and the bottom electrode, 25 PMUTs vibrate at the same time, and the ultrasonic waves generated by all the vibration elements are superimposed and then emit ultrasonic waves to the external medium, a 5 × 5 array element type structure. The structure can greatly increase the energy of ultrasonic waves emitted by the PMUT device; when the PMUT is used as a receiving device, the electrical displacement caused by the vibration of 25 PMUTs is superimposed on the extraction electrode, thereby improving the signal output capability of the device.

The PMUT is simulated using COMSOL multiphysics 13.0 software. Figure 3a shows the mode shape of the 5 × 5 PMUT arrays at the resonant frequency. All elements have consistent mode shapes. Figure 3b shows that the output acoustic wave of a single PMUT is omnidirectional, and Figure 3c shows the *Z*-axis directivity of the PMUT arrays. The directivity of the PMUT arrays in the *Z*-axis direction is much higher than the sound pressure intensity in any other direction.

This article uses MEMS processing technology to prepare the designed PMUT device. The key preparation process flow is shown in Figure 4. The preparation process is described as follows: Select an SOI wafer composed of a 5 μm silicon device layer, 1μm buried silicon oxide, and a 500 μm silicon substrate. Physical vapor deposition technology was used to deposit a 100 nm buffer layer, 100 nm pt bottom electrode, 1 μm PZT piezoelectric layer, and 100 nm Pt top electrode on the SOI wafer. The top electrode is patterned using reactive ion etching (RIE). The bottom Mo electrode is exposed through ion beam etching (IBE) to form a through-hole design, which provides conditions for the electrical connection of the bottom electrode of the PMUT device. Deposit 200 μm of Au and form connecting lines to connect each micro-element in parallel, and deposit the pads of the PMUT array. Finally, deep reactive ion etching (DRIE) is used to etch the silicon substrate, and the etching of the back cavity accurately defines the diameter of the PMUTs, forming the vibrating membrane structure of the PMUT device.

### 3.2. Characterization

Figure 5 shows an optical microscope image of a 5 × 5 PMUT array. The dimensions of the fabricated 10 × 10 array are approximately 3.7 × 3.7 mm^2^. A total of 25 PMUT micro-elements are connected through Au wires to form a parallel array element structure, and the diameter of the Pt top electrode is 364 µm. The piezoelectric thin film device was cut longitudinally and SEM electron microscopy scanning was performed on its cross-section. The results are shown in Figure 6. From the figure, you can see that the diameters of the five longitudinally arranged PMUT element cavities are 518 μm, 525 μm, 520 μm, 523 μm, and 518 μm, respectively, which is not much different from the designed 520 μm cavity diameter, indicating that the process consistency is high and ensures the high performance of the device.

The mechanical characteristics of PMUT devices are specifically reflected in their vibration characteristics. In order to analyze the vibration characteristics of PMUT devices, laser Doppler vibration (LDV) testing was performed on the PMUT vibration element as follows: Set the scanning frequency range to 100–150 kHz and apply a sine wave signal with a peak value of ±5 V to the PMUT device. The measurement results are shown in Figure 7. The resonant frequencies of different vibration elements have certain deviations compared with the theoretical design. After testing and analyzing the chip, it was found that this deviation was caused by the MEMS process. There are several main reasons. First of all, although the buried oxide layer of SOI is selected as the cutoff layer for back cavity etching, the large depth-to-width ratio of the back cavity results in poor consistency in deep silicon etching, resulting in uneven thickness of the vibration film. Secondly, this design achieves stress relief by patterning the film, which may cause uneven stress release in the film, thereby affecting the vibration state of the device. Therefore, there will be a certain deviation in the resonant frequency of devices of the same design, but the overall resonant frequency point range remains at 120–130 kHz, based on which the passband of the filter circuit in the signal conditioning circuit can be determined.

To test the electrical characteristics of the designed PMUT device, extract its impedance and phase, and then analyze its electromechanical coupling coefficient. Use a vector network analyzer to test the electrical characteristics of the PMUT device, set the scanning frequency to 100–135 kHz on the network analyzer setting interface, scan and test its scattering parameter (S) parameter, and convert the S parameter into an impedance characteristic curve. The test results are shown in Figure 8. The series resonance frequency of the device is *f*_s_ = 122.85 kHz and the parallel resonance frequency is *f*_p_ = 125.82 kHz. The effective electromechanical coupling coefficient $k_{t}^{2}$ ≈ 5.68% of the PMUT device designed in this article was calculated. It shows that the acoustic transducer device designed in this article has good electromechanical performance.

### 3.3. Transmission Performance

The manufactured PMUT device was tested for ultrasonic time-of-flight determination. An ultrasonic echo test system was built as shown in Figure 9. The transmitter device PMUT A was connected to the output cable of the signal generator, and the output signal of the receiver device PMUT B was introduced. Next, amplify the filter circuit board; the signal generator provides the device with a sinusoidal AC signal with a frequency of 125 kHz and an amplitude of ±5 V, and the oscilloscope displays the excitation signal and the conditioned PMUT signal.

As shown in Figure 10, the red part is the 125 kHz sine wave excitation signal TX. Under the excitation of TX, a clear and stable ultrasonic signal RX at the receiving end can be observed. In the first few signal periods, the amplitude of the ultrasonic signal RX is relatively large. Low, with continuous signal excitation, the amplitude gradually increases, reaching a maximum value of 2.4 V at the tenth signal peak, and then gradually attenuates. The time difference Δ*t* between the excitation signal and the ultrasonic signal is the ultrasonic time-of-flight measurement result. Changing the separation distance between PMUT A and PMUT B will also change the TOF.

Since the front-end signal amplitude of the received ultrasonic signal is low and is easily affected by interference signals, it is relatively difficult to directly detect TOF. Select the fifth peak of the ultrasonic signal as the detection object, and set the corresponding voltage detection threshold according to its voltage amplitude. Then, compensate the measurement time according to the sinusoidal period of the ultrasonic signal to obtain the ultrasonic flight time. The results show that the PMUT device produced in this article can complete the emission of ultrasonic signals under corresponding signal excitation; combined with the amplification filter circuit, the PMUT device can receive ultrasonic waves near its resonant frequency point and obtain a clear and complete ultrasonic voltage signal.

## 4. Experiment Results and Discussion

As shown in Figure 11, the circuit system mainly consists of two PMUT acoustic transducers, a transceiver switching circuit, a signal receiving module, a signal driving module, and an FPGA timing and logic control module. Combine the PMUT device with the signal conditioning circuit and the FPGA circuit to build an ultrasonic gas flow measurement system. After the system is powered on, under the control of the counting and time difference measurement module, the DDS signal generator implemented inside the FPGA generates digital signals containing drive signal information. The value is then sent to the DAC module through the FPGA external interface. The DAC module then converts the digital value into an analog voltage signal with an amplitude of ±5 V and a frequency of 125 kHz to excite the transmitter PMUTs and drive the transmitter PMUTs to transmit ten ultrasonic signals. After a period of time, the ultrasonic signal is received by the receiving end PMUTs. After amplification and filtering, an ultrasonic voltage signal that is easier to identify and detect is obtained. The ultrasonic voltage signal is then converted into a digital signal through the ADC module and transmitted to the FPGA. The internal control module obtains the ultrasonic flight time. Complete the time-of-flight measurement in the downstream stage; after the downstream time-of-flight measurement is completed, switch the sending and receiving relationship, repeat the measurement, and obtain the time-of-flight difference at the current flow rate. After calculation, the current gas flow rate in the pipeline is obtained.

This paper uses a pipe section where the fluid path and the acoustic path coincide with each other to measure the gas flow rate. The PMUT acoustic transducer is fixed on the PCB with silicone, bound to the PCB pad through gold wire, and then fixed at both ends of the acoustic path. The distance between the two ultrasonic transducers is 100 mm, and the diameter of the 3D-printed fluid pipe is 20 mm. The flow test was conducted at a temperature of 15 °C and a humidity of 50%. The speed of sound under this temperature and humidity condition is 341 m/s. When the fluid medium is stationary, the time-of-flight of the ultrasonic wave from transmitting to receiving is approximately 294.118 µs. Every time the medium flow speed increases by 0.1 m/s, the time-of-flight difference between downstream and countercurrent increases by about 250 ns. The ultrasonic gas flow measurement system is shown in Figure 12.

After the ultrasonic flow measurement system is built, the gas flow in the pipeline is changed and measured multiple times. By comparing the downstream flight time and the countercurrent flight time, the ultrasonic time-of-flight count difference was obtained. Finally, the count difference was converted into a time difference and the current measurement value was calculated from the data. A standard anemometer (Xinste HT9829, Dongguan, China) was used for synchronous measurement, and the measured value was used as the standard value. The resolution of the anemometer is 0.01 m/s, the measurement range is 0.1–25.0 m/s, and the accuracy is 1% of the anemometer range.

Thirteen groups of flow rates were selected in the range of 0–7 m/s, and each group of flow rates was measured six times. The measurement results are shown in Figure 13. This repeated experiment found that there is a certain error between the fluid flow rate measured using the flow measurement system built in this article and the standard value. This error floats within a certain numerical range. Through analysis, it is believed that this error is caused by circuit delay and PMUT installation accuracy.

Since PMUTs are not chip-level packaged, the devices are easily disturbed by the external environment, which in turn affects the measurement accuracy and measurement range of the system. There is still room for improvement in the measurement accuracy of the designed flow measurement system. According to the flow measurement principle of the transit time method, it can be determined that the fluid flow rate in the pipeline is proportional to the count difference. Based on the obtained data, a first-order linear fitting is performed on the flow rate-count difference, and the result is shown in Figure 14. It can be seen from the fitting curve in the figure that there is a good linear relationship between the gas flow rate in the pipeline and the count difference recorded by the FPGA, which is in line with the experimental expectations.

After compensating the error, retake the measurement point within the measurement range for measurement, and obtain the relative error of the measurement results as shown in Figure 15. Within the measurement range of 0.5–7 m/s, the maximum error is 0.17 m/s, and the maximum relative error is 4%.

Based on the above experimental results and analysis, this system implements an ultrasonic flow measurement system based on PZT thin film PMUTs. The parameter comparison with other flowmeters of the same type is shown in Table 1. The developed MEMS gas flowmeter shows the advantage of miniaturization in terms of pipe diameter. A gas flow velocity measurement range of 0.5–7 m/s was achieved. The research results have a certain value. However, due to the impact of packaging and installation, the accuracy of the system still needs to be further improved. And subsequent research on ASIC circuits can further optimize the errors caused by the circuit.

## 5. Conclusions

In summary, this article takes the PZT piezoelectric ultrasonic transducer as the core and describes a small diameter gas flow system based on the time difference method: the final prepared transducer array consists of 5 × 5 PMUT microelements. Composed, the resonant frequency of the array is about 125 kHz, the size is 3.7 × 3.7 mm^2^, and the electromechanical coupling efficiency is about 5.68%. Taking the prepared PMUT device as the sensing core and using the time difference method gas flow measurement as the application design circuit system, including signal receiving circuit, signal transmitting circuit, FPGA design, etc., combine the system circuit and oscilloscope and other instruments to conduct flight time verification of the PMUT device. The experiment and test results show that the prepared PMUT device has good transceiver performance. Finally, an ultrasonic gas flow measurement system was built to achieve high-precision measurement of gas flow in DN 20 pipe diameter. The flow rate measurement was 0.5–7 m/s, the maximum error was about 0.17 m/s, and the relative error was about 4%. Future work will focus on volumetric flow measurement for smaller pipe diameters.

## Figures and Tables

**Figure 1 micromachines-15-00045-f001:**
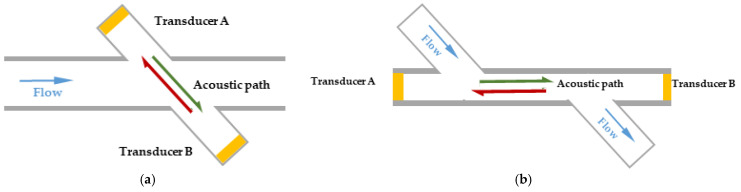
(**a**) Principle diagram of traditional time difference method for flow measurement; (**b**) schematic diagram of structural improvement.

**Figure 2 micromachines-15-00045-f002:**
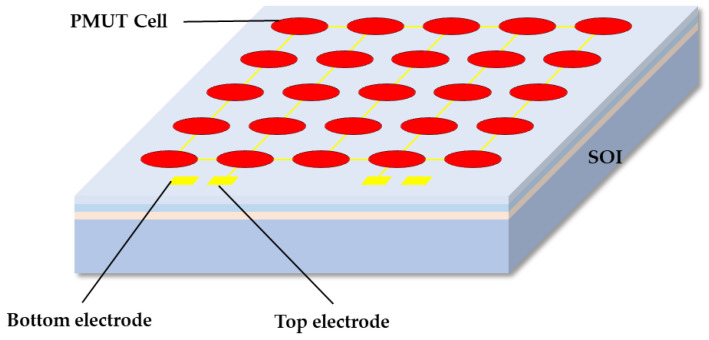
The 5 × 5 array element PMUT structure.

**Figure 3 micromachines-15-00045-f003:**
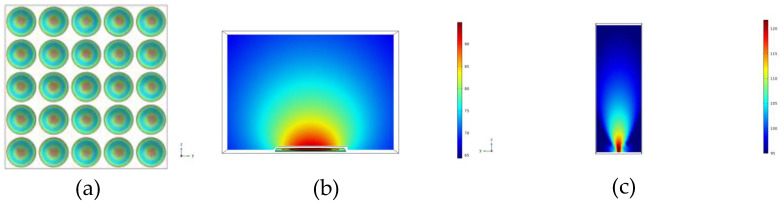
(**a**) Vibration mode shape of the PMUT arrays. (**b**,**c**) The sound field distributions of a single PMUT array and a 5 × 5 PMUT array, respectively.

**Figure 4 micromachines-15-00045-f004:**
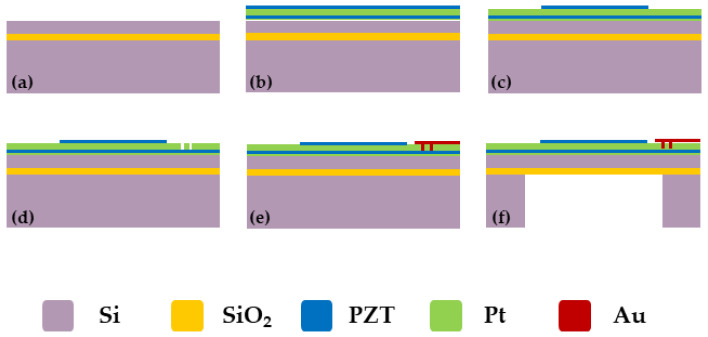
PMUT processing process flow. (**a**) Select an SOI wafer; (**b**) Growth electrodes and piezoelectric layers; (**c**) Patterned top electrode; (**d**) Graphic electrode outlet hole. (**e**) Grow and pattern leads and pads. (**f**) Graphical back cavity.

**Figure 5 micromachines-15-00045-f005:**
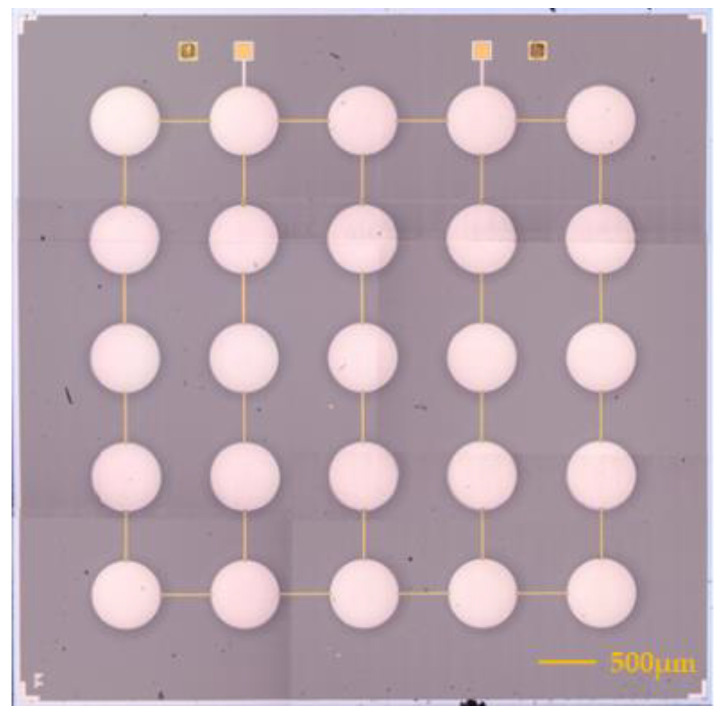
The image of optical morphology of PMUTs.

**Figure 6 micromachines-15-00045-f006:**
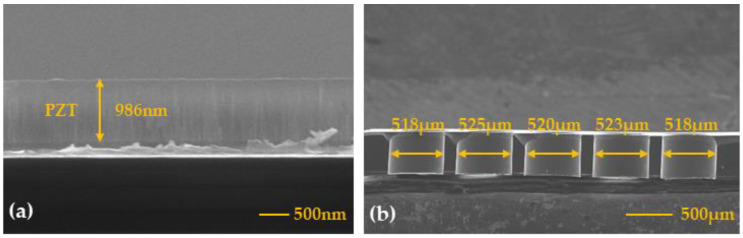
The SEM image of the cross-sectional view. (**a**) The SEM image of the fabricated PZT thin film. (**b**) SEM image of the etched cavity.

**Figure 7 micromachines-15-00045-f007:**
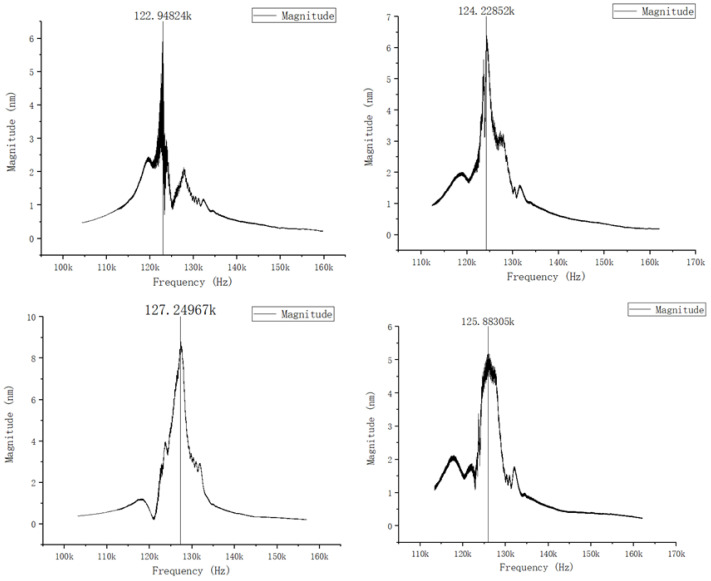
Amplitude–frequency curve of PMUT element.

**Figure 8 micromachines-15-00045-f008:**
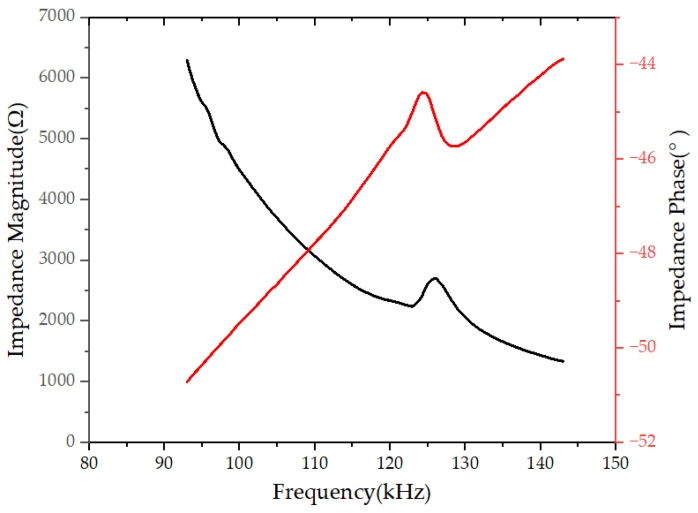
Impedance curve of PMUT device.

**Figure 9 micromachines-15-00045-f009:**
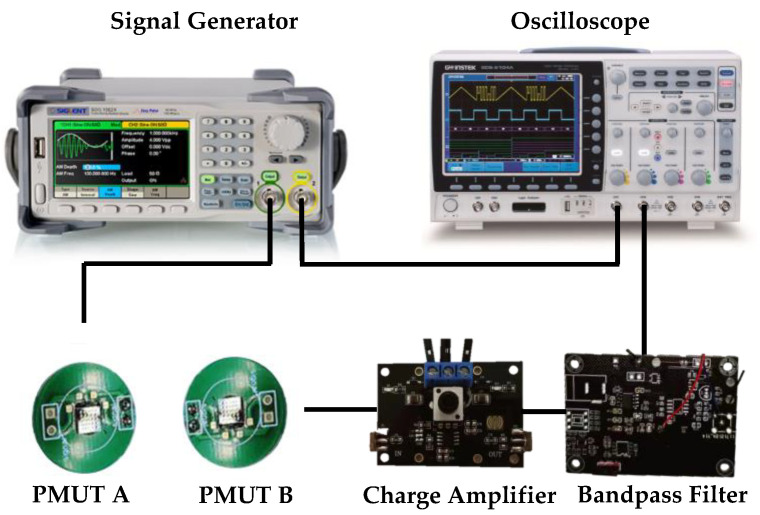
Ultrasonic time-of-flight test system.

**Figure 10 micromachines-15-00045-f010:**
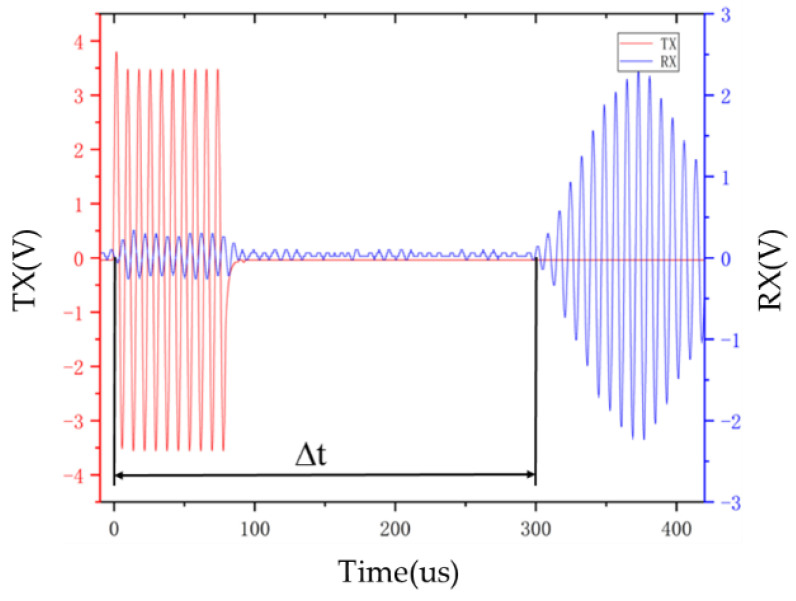
The ultrasonic signal received during interval excitation.

**Figure 11 micromachines-15-00045-f011:**
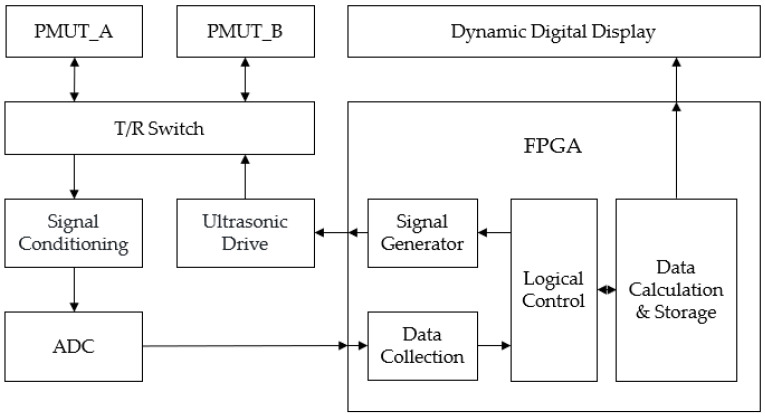
Circuit system block diagram.

**Figure 12 micromachines-15-00045-f012:**
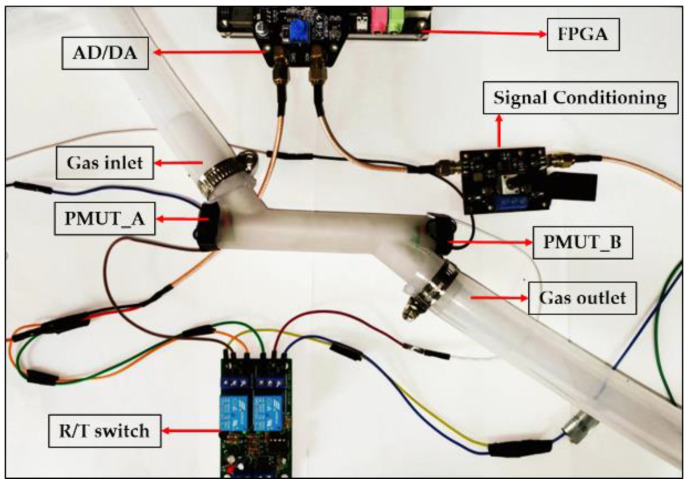
Ultrasonic gas flow measurement system.

**Figure 13 micromachines-15-00045-f013:**
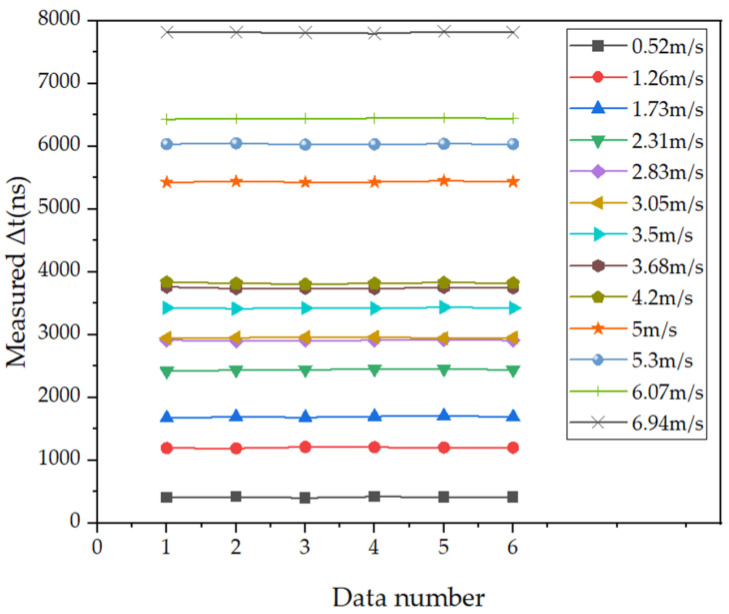
The eight measurement results of time difference for different flow rates.

**Figure 14 micromachines-15-00045-f014:**
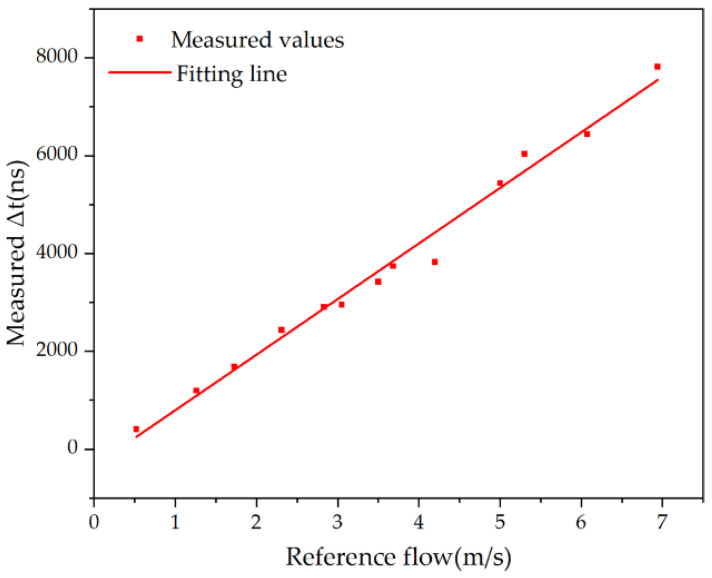
Reference flow rate and time difference fitting results.

**Figure 15 micromachines-15-00045-f015:**
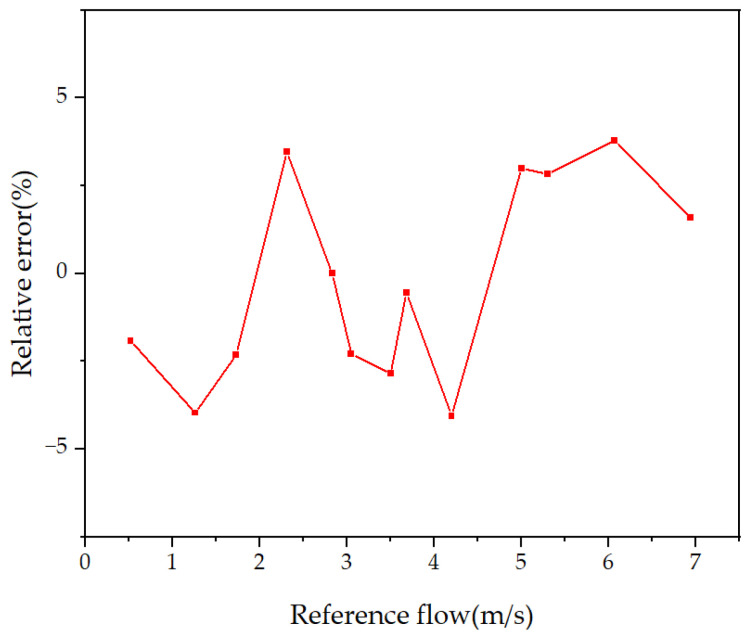
Relative error of gas flow measurement.

**Table 1 micromachines-15-00045-t001:** Parameter comparison with other flowmeters.

	Fluid Type	Measurement Range	Absolute Deviation	Relative Deviation
This work	Gas	0.5–7 m/s	0.17 m/s	4%
Ref [21]	Gas	0–2 m/s	0.1 m/s	5%
Ref [22]	Liquid	0.17–1.13 m/s	/	3%
Ref [24]	Liquid	0–0.28 m/s	/	5%
Ref [29]	Gas	0.15–3.2 m/s	0.12 m/s	3.9%

## Data Availability

The data presented in this study are available on request from the corresponding author.

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
