# Peer review of "A Gas Flow Measurement System Based on Lead Zirconate Titanate Piezoelectric Micromachined Ultrasonic Transducer"

_micromachines, 2023, doi:10.3390/mi15010045_

Round 1
Reviewer 1 Report
Comments and Suggestions for Authors
The authors report on an ultrasonic flow sensor consisting of two 5x5 PMUT arrays. Although the authors explain the principle of the fabricated and characterized device quite well, there are some major concerns about the novelty and scientific value of this work.
1. The introduction gives a good overview of ultrasonic flow sensors. However, the authors do not clearly position their research in the field. What is the novelty (gap in knowledge)? Is the author's method e.g., smaller, more accurate or has a higher range? Or is it specifically that you use PMUT arrays in stead of single transducers? In Table 1, it appears that the sensor has a significantly higher range compared to other work, is this maybe the most important achievement?
2. Why did the authors choose for a 5x5 array of PMUTs? The reasons given are nog very quantitative and do not form complete reasoning. Find below some examples of questions that need to be answered in Section 3.
- Why a 5x5 array and not less bigger transducers?
- Since you are using arrays, can you say anything about phased array or beamforming techniques to improve this method.
- Why is it a 5x5 array and not a 25x1 array (or anything in between)?
- 'greatly increase the energy', how much compared to a single element?
- What are the downsides of a 5x5 array? Why not always use 5x5 arrays in ultrasonic transducers?
The same holds for the tube shape (acoustic path), its angle and any other design choices: elaborate more on how the optimum has been found.
3. It has been claimed in the introduction that ultrasonic flow sensors have a high precision, but no statistical analysis on the data has been performed. Figure 13, for example, does not have any error bars. For each applied flow, could you measure multiple points and find the standard deviation or other measure for error?
4. Minor corrections:
- Line 19, 299, 323: '~' should be '-' or 'to'
- Line 24-26: be careful with words like 'always' and 'important'
- Line 38: The -> the
- Line 150: 'this article designs'
- Line 155: what is a through hole?
5. Typesetting of equations and variables could be significantly improved for readability.
- use italic for in-line variables
- operators like 'cos' should be non-italic
- subscripts should be non-italic
- use large brackets for divisions: e.g., line 123 and 127
- subscript missing (e.g., tup should be t_up on line 125 and fp on line 222)
- Line 123: t_dowm -> t_down
Reviewer 2 Report
Comments and Suggestions for Authors
The manuscript presents a pMUT for flow measurement its characterization experiments in details. I have the following comments:
1. The introduction doesn't give sufficient introduction of pMUTs and relevant references. The current challenges and novelty of this work are not clearly stated.
2. From the perspective of the piezoelectric material, pMUTs have been fabricated based on sputtered AlN, sol-gel PZT, sputtered PZT and thin ceramic PZT. The manuscript doesn't give these introduction, what type of PZT is used in this work?
3. Need more details for the design strategy, eg., why is 125kHz selected as the resonant frequency? What is the consideration to design 5x5 elements?
4. Several typos. e.g., line 73-74.
Comments on the Quality of English LanguageMinor editing
Round 2
Reviewer 1 Report
Comments and Suggestions for Authors The reviewers have written a clear response to the concerns. The positioning of this research in the introduction has been made much stronger. On the other two major issues, the authors have written a decent response, but these have not been implemented in the paper.- The authors have written a strong analysis of phased array in their cover letter, but did not include this reasoning behind the design in the paper. Section 3 (design) would benefit from the information given in Response 2.
- Thanks for the table with the raw data. It is unclear why the provided data in the response (13 flows, 5 points per flow) has been used for Figures 13 and 14 and not the data from Figure 12 (6 flows, 8 points per flow). Or the other way around, why the provided data in the response has not been used for Figure 12. Best would be to make a decision here and use one dataset for all plots to improve clarity.
- Figure 13 contains 13 points, Figure 14 misses the last point (contains 12 points). Please include the last point or explain why this has been left out. Note that your claim of the maximum error is based on the range up to 7 m/s, so excluding this point is certainly not allowed.
- Line 308-315 - use of imperative, please use a different grammatical tense. E.g.,: “convert the count different into” -> “the count difference has been converted into”.
- Line 165-166 - not sure how to read this sentence; incorrect grammar or typo.
- Line 656: “systematic error” is not a sentence.
- The authors have written a strong analysis of phased array in their cover letter, but did not include this reasoning behind the design in the paper. Section 3 (design) would benefit from the information given in Response 2.
- Thanks for the table with the raw data. It is unclear why the provided data in the response (13 flows, 5 points per flow) has been used for Figures 13 and 14 and not the data from Figure 12 (6 flows, 8 points per flow). Or the other way around, why the provided data in the response has not been used for Figure 12. Best would be to make a decision here and use one dataset for all plots to improve clarity.
- Figure 13 contains 13 points, Figure 14 misses the last point (contains 12 points). Please include the last point or explain why this has been left out. Note that your claim of the maximum error is based on the range up to 7 m/s, so excluding this point is certainly not allowed.
- Line 308-315 - use of imperative, please use a different grammatical tense. E.g.,: “convert the count different into” -> “the count difference has been converted into”.
- Line 165-166 - not sure how to read this sentence; incorrect grammar or typo.
- Line 656: “systematic error” is not a sentence.
Round 3
Reviewer 1 Report
Comments and Suggestions for Authors
The authors have significantly improved the paper and can therefore be accepted if there are no further concerns by any other reviewer.